# A Methodological Approach to the Teaching STEM Skills in Latin America through Educational Robotics for School Teachers

**Sandra Cano** 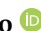

School of Computer Engineering, Pontificia Universidad Católica de Valparaíso, Valparaíso 2340000, Chile;
sandra.cano@pucv.cl

**Abstract:** The study aims to design a methodological approach that allows educational robotics to develop STEM competences for schoolteachers, but with a gender focus. The phases within consist of designing a set of workshops with a gender approach, making use of Arduino, as it allows for introducing concepts in electronics and programming. For this, a mixed research method was applied, where quantitative and qualitative information was collected. The study was carried out with teachers from Latin American schools, where teachers from Chile and Colombia participated the most, and was conducted in virtual mode through the Zoom platform. As a result, it was found that Arduino and its components can be used to build projects that can be related in a real context, which further motivates students. It was also found that the levels of creativity, attitude, and motivation of the students increased with the workshops that were carried out.

**Keywords:** educational robotics; computational thinking; STEM

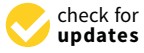



## 1. Introduction

Young girls are generally taught to knit while boys are taught to make wooden boats. Children thus associate various activities with a particular gender. Gender roles, in turn, are culturally stereotyped behaviours. They are thus activities that a person is expected to perform according to his or her gender [1].

Nowadays, gender disparities are especially pronounced in areas such as computer sciences and electronics, female sign-up remains low. Therefore, differential experiences in STEM continue for women and men at the high school level. Some authors indicate inadequate early preparation is problematic for women in computer sciences [2]. Therefore, low participation in STEM courses can limit their ability to access STEM careers later. Margolis et al. [2] found that women lost confidence and interest in computer science because they felt they did not fit with the stereotypical view of a computer scientist. Therefore, women's decisions are very much subject to those barriers arising from basic education, because they do not see themselves identified or feel similar enough to those scientists and computer and/or electronic engineers to enter in these fields [3].

When they do enter the fields in question, they can be penalized socially and professionally for exhibiting leadership skills and qualities [4]. Therefore, some of these barriers contribute to why women choose to enter other fields and lose interest in careers such as electronic and computer science.

In many countries, girls' education is considered an essential element for economic development. Initiatives such as "Roberta initiative" [5] used robot construction kits in combination with gender-balanced didactic material and course concept for girls' interest in technical topics. Another example is WSTEM [6], an Erasmus project to promote STEM careers for women in Latin America. The Girls4STEM project [7] works towards breaking the stereotypes linked to STEM fields, addressing both boys and girls aged from 6 to 18, but especially young girls through interaction with female STEM experts. Therefore, there

is great interest in promoting the participation of girls in STEM from an early age, so that women can become more involved in engineering careers and reduce the gender gap, especially in engineering careers such as computer science and electronics.

However, in Latin America. this may not always be the case, and legal, institutional, political, and cultural aspects of their environment mean that many women and girls in the world are excluded from science and technology activities [8,9]. A report presented by UNESCO [10] spoke of the movement taking place in different institutions to promote science, technology, and gender issues. More than 1 billion people live in poverty in Latin America, the majority of whom are women and children; the role of science and technology in society has become vital for improving the quality of life and the socio-economic and environmental situation of any country.

A report published at UNESCO in 2015 [10] mentions that 58% of women in Latin America tend to earn less and are in minority in fields such as sustainable development, information technology, and computer science. In the same document, they also presented a report by country of female researchers in technology, where in 2012 Colombia represented 21.6%, in 2008 Chile had 19%, in 2011 Costa Risa had 30.9%, in 2013 El Salvador had 17.7%, in 2012 Guatemala had 43.5%, and in 2009 Venezuela had 40.4%.

Today, women still suffer from low participation in STEM areas, not only as students, but also as teachers, researchers, and workers [11]. Different factors frame the gender gap. Traditionally, it has been thought that boys have more talent for mathematics and technology and girls have more talent for verbal skills [12]. Studies show that stereotypes associated with technology, physics, and engineering negatively influence girls. However, schools still do not have teachers who are technologically and pedagogically prepared in the areas of STEM, especially in the technology and engineering fields [13]. Therefore, if boys/girls are educated in STEM areas from an early age, one can help gender inclusion from that moment.

Educational robotics is a growing interest in STEM at all levels, especially to promote STEM careers for women. The use of a robot in programming education could help girls understand computer sciences concepts. However, educational robots have a more significant effect on boys than girls [14]. Zhang et al. [14] indicated than there is a "negative stereotype" for girls, which causes them to feel less able to study STEM. Meanwhile, boys are traditionally more familiar with the technology. Educational robotics (ER) is usually seen as an interdisciplinary activity in science, technology, informatics, and mechatronics. Therefore, ER is a powerful, flexible teaching and learning tool to construct robots and control robots using tangible programming languages [15]. Furthermore, ER activities help students become active learners. However, girls appear to need more training time in many situations to reach the same skill level compared to boys [16].

A study conducted by Sullivan and Bers [17] explored the gender differences in student experiences in robotics competitions. Some observations were as follows: "females tend to stand back and let the males take the lead in building even if the males don't know any more about the task", "most of the girls were not as inclined to want to actually build something. They have to be encouraged to use a wrench", "females at my school have had less experience at constructing so they feel insecure or just do not know how to put things together to make what they want". Sullivan and Bers identified that one reason female students may be less confident in their technical and building skills is that female students may simply have less experience with building, tinkering, and constructing prior to joining a robotics competition in middle or high school teams. Research has shown that women have less experience with tinkering during their childhoods compared to men [18], which can be influenced according to the stereotypes they are exposed to and according to interest. In 2018, Sullivan and Bers [19] examined the impact of girls having females as robotics teachers. The study was conducted with female teachers using a prototype of the KIBO robotics kit, which was designed for children aged 4–7 years. The tasks tested were sequencing, a repeat loop, and a conditional statement with two levels, easy and hard,

based on how many commands children needed to sequence, with fewer blocks for children to sequence than hard tasks.

In 2020, Román-Graván et al. [20] carried out a study related to perception in the use of educational robotics in training for future teachers. In the study, they performed several robotic kit interventions, such as Colby robot mouse, Ozobot, mBot and the Makey-Makey board. To learn the programming language they used Scratch, where they had to design a video game related to healthy eating using a tangible interface such as the Makey-Makey board. Once they interacted with these kits, the authors applied an instrument that consisted of 42 items that enabled an understanding of the perception of the use of robotics, in which they expressed their motivation to implement it within the subjects. They further indicated that educational robotics in the classroom could promote new teaching–learning methodologies for students and favour the development of self-learning skills.

In order to enhance educational robotics, Peixoto et al. [21] used Raspberry Pi and Arduino as the hardware interface. Cuartielles et al. [22] introduced robotics concepts using an Arduino-like tool. A study conducted by Ntourou et al. [23] used Arduino and Scratch to study their effect on self-efficacy and motivation towards science education and computational thinking in 5th grade students about concepts of electricity. Abidin et al. [24], to promote STEM education learning, designed a process of educational robotics for teachers involved in designing and constructing robots using open source and low-cost technologies such as Arduino. The studies reviewed use robotics in education to promote computational thinking. However, the introduction of concepts of electronic and mechanical parts to build a robot is not considered.

On the other hand, methodologies to teach robotics with robots is not clear. Some studies such as Dimitriou [25] propose a methodology that follows seven steps, such as teaching theory, teaching tools, problem selection, analysis, design, implementation, and evaluation. O. de Azevedo et al. [26] present a methodology composed OF five steps, namely the initial diagnosis, survey of contextualized problems, course planning, classes, and a robotics fair. However, these methodologies do not have a gender approach.

Therefore, the main objective of this study is summarized in the following research questions: What aspects should be considered to propose a methodology with a gender approach to motivate women to choose studies related to engineering, especially computer sciences and electronics?

## 2. Background

### 2.1. Educational Robotics

Educational robotics (ER) is a sub-field of robotics that provides students with learning experiences through the creation and implementation of activities, technology, and artifacts related to robots [27]. Educational robotics began with the Logo project developed by Seymour Papert [28], a mobile robot in the shape of a turtle [29] to teach programming to children [30]. The turtle could be programmed to draw pictures on the surface on which it moved using a pen that was in the bottom center of the robot.

Educational robotics has mainly focused on supporting the teaching of subjects that are closely related to the robotics field such as programming, construction, and mechatronics. However, the studies found have used the robot as a passive tool in which students must program the robot. Rush et al. [31] mention that students who are not interested in traditional approaches become motivated when robotics activities are introduced as a way to tell a story or in connection with other disciplines and interest areas. A report from the American Association of University Women [32] argues "girls and other nontraditional users of computer science-who are not enamored of technology for technology's sake-may be far more interested in using technology if they encounter it in the context of a discipline that interest them".

Therefore, robotics construction kits can be used in many different ways, to support many types of activities and different learning styles. Plaza et al. [33] used the Arduino embedded system as an educational tool to introduce robotics, where children built

and developed tangible prototypes for problem-solving. PicoCricket [31] is a robotics kit that aims to combine art and technology, enabling young people to create artistic creations. PicoCricket has output devices such as motors, colored lights, and music-making devices and sensors. In 2019, Xenabis et al. [34] made use of recyclable materials and programming with Arduino UNO, where they built the Wall-E robot and programmed the robot through a platform called Ardublock. Another work was carried out by Junior et al. [35] in which they proposed a low-cost educational robotics kit based on the Arduino UNO platform. To design the robotics kit, four requirements were considered: Low-cost, appeal, simplicity, and opensource. For the programing environment, they used block programming called a mini block. For the use of this kit, the following eight learning modules were designed: (1) What are we going to learn? (2) What is robotics? (3) What is Arduino? (4) Learning to program with Minibloq; (5) Electronic components; (6) What are sensors? (7) Robot architecture; and (8) Robot operation. At the end of the course, they asked related questions about whether the kit was a good option for understanding the concepts of electronics and programming.

Educational robotics has in turn been associated with the field of computational thinking [36] and related to STEM learning [37]. Today, educational robotics is being included in the classroom as a form of teaching–learning that can help the development of competencies and promote learning in areas such as engineering, technology, mathematics, and science. Several studies have shown that educational robotics has a positive impact in STEM areas [38–40], as it promotes an understanding of STEM-related concepts. ER can be effective in teaching STEM [41] because it allows one to interact with the real-world concepts of engineering and technology.

Nowadays, ER is being implemented in schools as an alternative to empower students in various related areas in engineering, science, and mathematics [42,43]. Teachers have started to develop activities to incorporate robotics into teaching. However, there are more individual initiatives.

Mataric [30] states that "robotics has the potential to significantly impact the nature of science and engineering education at all levels". In turn, educational robotics began to be used in robotics competitions as a way to encourage their learning. These contests even employ goal-oriented and project-based learning (PBL), and the contests are geared mostly towards the engineering, computer science, and artificial intelligence fields.

On the other hand, Barreto and Vavssori [44] mention that ER is related to thinking skills, the scientific process, problem-solving approaches, and teamwork skills. A study presented by Alves-Oliveira [45] features activities that enhance creativity in children. For this study, they carried out three activities. The first activity was to code the robots. The second activity involved learning to design robots, while in the third activity condition, a control participated in a music class. In the first activity of this study, they learned to use Scratch language.

## 2.2. Gender in Educational Robotics

Sapounidis et al. [46] found that girls have strong preferences for tangible interfaces, and programming-related tasks can be more difficult for them, due to the manner of teaching. A study conducted by Blue and Gann [47] mentions that girls start kindergarten interested in areas such as math and science but leave high school with that interest far diminished. Therefore, girls lose interest in science and mathematics as they go through school, specifically from fourth grade. Furthermore, girls and women often receive the message that the fields of science, technology, engineering, and mathematics are not for them [48]. Another study [49] examined interactions with a formal educator where they observed that girls are less concerned about being negatively stereotyped when their teacher is female than when their teacher is male. Studies [50] have shown that girls and women are more interested in careers where they can help others.

Sullivan and Bers [43] found differences between girls and boys, where girls tend to back off and let boys take the lead in construction even if boys do not know much about the

task. They also found that both boys and girls are good at construction, but boys are more likely to take control or lead. Therefore, girls tend to be more passive when it is a mixed team. At the same time, the girls often do not take control since they are afraid that they will be made fun of by their male companions. However, even the preferences and the correct way to teach girls educational robotics have not been sufficiently researched [51].

### 2.3. Arduino-Assisted Robotics Coding Applications

Arduino is a microcontroller card created by Massimo Banzi in 2005. This card constitutes easy-to-use hardware and software based on an open-source electronic platform [52]. Arduino allows a wide range of applications, from robotics to automatic control systems. Arduino can be programmed with block-based coding such as the mBlock coding platform, scratch, and TinkerCad. The ability to add advanced technologies to these boards plays an important role in the use and dissemination of Arduino-assisted robotic coding applications in educational environments [53].

Arduino is a card that can handle both analog and digital signals. It integrates a variety of communication protocols such as SPI (Serial Peripheral Interface), I2C (Inter-Integrated Circuit), serial communication, and UART (Universal Asynchronous Reception Transmission). Arduino allows students to control the reactions of a system that they can visibly touch and see and makes it possible for learners to problem solve in situations they encounter in everyday life. This idea is due to the fact that Arduino can be integrated with various sensors such as temperature, humidity, speed, sound, light, gas, color, vibration, and distance, among others. Therefore, with the use of sensors Arduino can sense what is happening around it, which allows a control or monitoring system to be developed. The use of sensors allows a great deal of interaction with science and engineering.

In the literature reviewed, Arduino-assisted robotic coding applications facilitate the teaching of abstract and difficult-to-understand concepts in science subjects, and such applications should be included in the teaching of science subjects such as medical science to monitor or simulate heart rate, detect body temperature, electricity, sound sensors that work with sound waves, etc.

On the other hand, learning Arduino involves many technologies depending on how far you want to go. Therefore, it can help increase the interest, attitudes, and motivation towards technology applications and science teaching.

### 2.4. Methodologies for Teaching ER

The term educational robotics is used a lot in schools. There is still no clarity on how the teaching of educational robotics ought to be, and especially from a gender perspective. Some methodologies have been proposed for learning, such as the work proposed by Patiño-Escarnina et al. [54], whose main objective is to understand how to introduce the concepts of robotics and related topics into the student curriculum. The authors focus on fields such as mechanics, electronics, control, and computing. The methodology they propose is made up of three phases: (1) Setting up the environment where a problem is defined and topics are selected; (2) definition of the project, where concepts and strategies are developed; and (3) conducting the assessment, where theoretical concepts are applied, and competencies are assessed. For the evaluation of competences, they include four variables: Communication, teamwork, creativity-responsibility, and integration of STEM topics. Based on Vygotsky's socio-cognitive approaches [55], activities involving educational robotics work through collaboration and teamwork.

O. de Azevedo et al. [26] proposed a methodology of contextualized ER, where it is necessary to start working by perfomirng a diagnosis at the school, with the students, and in their community. The methodology is composed of five steps, such as the initial diagnosis, survey of contextualized problems, course planning, classes, and a robotics fair. The methodology was proposed but not evaluated. Another study by Dimitriou [25] proposed a methodology of seven phases. During the first two phases, the teacher follows a predetermined pattern where the main objective is to explain theoretical concepts and

train students in software. All other phases require the learning process, so the teacher acts as a coach and cognitive modeler.

Barak and Zadok [56] described three strategies that lead to innovative solutions in robotics tasks—assign a new role (the students find a new use for the robot); remove a component from the system; and examine physical objects available in the environment and apply them to solve a problem. Several studies have demonstrated that educational robotics has a positive impact on the development of skills such as critical thinking [57], problem solving [58], metacognitive skills [59], and creativity [60].

Educational robotics continues to require more research, which would indicate how to work with educational robotics in order to develop skills in students, since these skills have not been evaluated in depth either. Sullivan [61] meanwhile identified that in the various stages in programming a robot, the students (1) write code, (2) test the robot, (3) analyze the problem, (4) propose changes to the model, and (5) test again. The author therefore identifies that the resolution of a problem involves three stages: (1) Identification of the problem, (2) generation of ideas and choice of strategy, and (3) reflection on the process of solving the problem.

Atmatzidou and Demetriadis [62] carried out 11 sessions to train students in robotics for public schools. They proposed a model to develop skills related to computational thinking within educational robotics. The authors focused on five dimensions of the conceptual framework of computational thinking: Abstraction, generalization, algorithm, modularity, and decomposition. They made use of the Lego Mindstorms NXT robot kit as a tool.

The studies reflect that there is no shortage of studies focused on educational robotics. However, the pedagogy of teaching robotics in schools is still in its infancy [63]. More research is therefore required on how to work with educational robotics for teachers in a way that can help students develop specific skills.

A study by [64] proposes a map of terms associated with educational robotics. Within the map of terms, the associated methodologies are project-based learning, problem-based learning, active learning, collaborative learning, experiential learning, and playful learning. All these methodologies are associated with constructivism and constructionism.

Today, many teachers remain unaware of the benefits of educational robotics and are still not prepared enough to be able to teach robotics or concepts involving educational robotics, such as electronics, programming, and technology. As such, there is a lack of specialized training programs for educational institutions focused on teachers since most of the studies found are focused on the student and not on the teacher. Some studies have ICT teachers as participants [65,66], while others feature STEM subject teachers [67]. A study in 2021 by [68] comprised a review of the literature on teacher training in educational robotics. The authors identified that the training programs include participants with different profiles, related to teaching experience, age, familiarity with technology, etc. In addition, many of the ER trainings present training programs without requirements, and those who have studies of programs with requirements have a final project of designing a robot, creating a program, or designing didactic material. However, many of the studies focus on building a robot, despite the fact that the majority of the teachers who enroll are teachers with a background in electronics or programming.

## 3. Methodology

In this research, a mixed research method was applied including both qualitative and quantitative data collection and analysis processes. The design of the research is given in Figure 1. Therefore, teaching STEM skills with a genre focus follows a set of phases proposed in Figure 1. Following a constructivist approach using a learning model called 5E (Engage, Explore, Explain, Elaborate, and Evaluate) [69], participants can experience meaningful instruction and learning for themselves within a practical, constructive, and active environment.

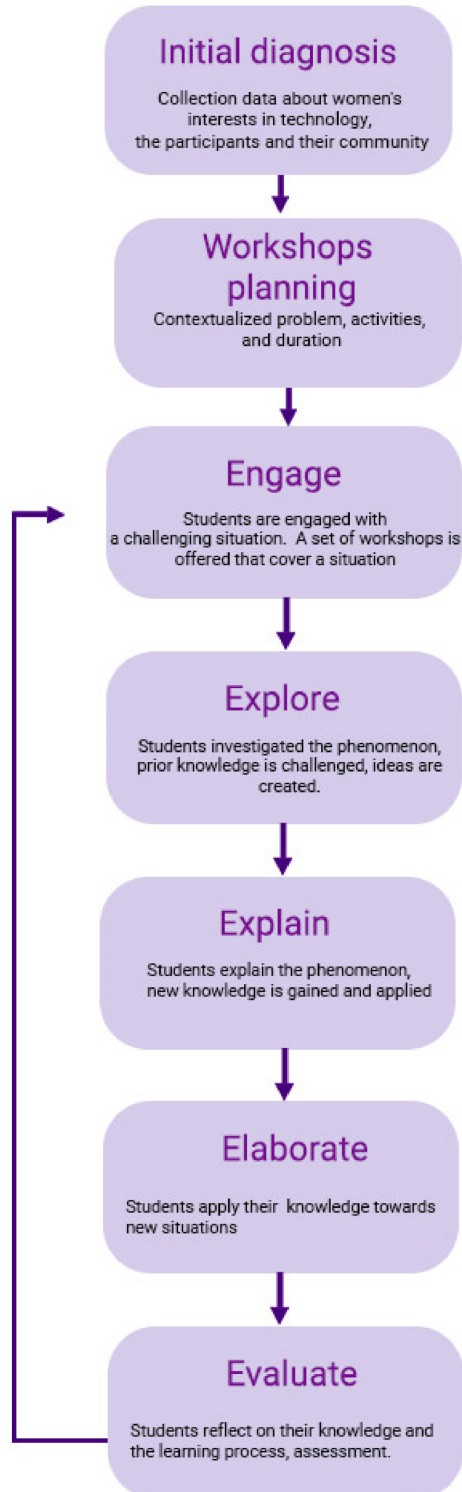

**Figure 1.** Phases proposed for Teaching STEM Skills in ER.

The first phase called the initial diagnosis collects data about women's interests, the profile of participants, whether they are schoolteachers, level of knowledge in technology, and their background in information technology and robotics. Demographic information is also considered.

The information obtained can help to gain initial insights into the interests and needs of women and the specific robotics/computing topics or curricular content to be further explored.

The second phase is related to initial ideas of possible themes addressing contextual problems. Therefore, the data collected in the first phase are considered. In this phase, it is necessary to propose a set of workshops that relates any topic = to the reality of where the students live, which can be approached with curricular and robotic content. The third phase engages students' prior knowledge about programming, components of electronics, and microcontrollers. The teacher starts the workshops with an interesting question about the subject. Therefore, the teacher shows possible ideas by encouraging students to participate and shows solutions that can be used with the use of technology and engineering. The fourth phase allows students to explore new knowledge through video-clips and lectures and support guides on how to develop the workshop. Moreover, it uses robotics materials such as the Arduino microcontroller, sensors, jumper cables, and other electronic components. The fifth phase involves the teacher creating a discussion environment in the group and in the classroom by asking the participants about the mechanism of how science can be taught with the use of programming and electronics. The sixth phase elaborates, whereby the teacher provides learners with opportunities to apply their acquired knowledge in solving problems. Finally, the seventh phase evaluates, and the students must build assemblies in each workshop, with the objective of understanding theoretical concepts and solving a problem in a specific context.

### 3.1. Study Group

The study of the research consists of schoolteachers and others interested in educational robotics in Latin America. The reason for the selection of the study participants is that there is still a low participation of women in STEM careers in Latin America. The objective is to promote the participation of girls and women in education robotics.

Two hundred and ninety people from different countries registered: Chile (124), Colombia (98), Ecuador (11), Mexico (26), Costa Rica (3), Peru (23), and Europe (5), where 47.5% of the registered correspond to men and 52.5% to women. Furthermore, 55.2% of those enrolled are college teachers, 23.7% students, 8.7% higher education teachers, and 9% other professions. However, the average number of participants who permanently attended each workshop was between 50 and 60, as they were not required to attend remotely. Workshops were recorded and uploaded to the platform, for those who found it difficult to meet the scheduled timetable.

### 3.2. Data Collection Tools

Attitudes towards technology is a scale developed by Cross et al. [70], which consists of 24 items and 5-point Likert type. The scale has four sub-dimensions: Learning desire (12 items), self-confidence (5 items), computational thinking (3 items), and teamwork (3 items).

A questionnaire was used to determine information about participants and interest about taking the course. A first part consisted of demographic data, and the second part referred to an open open-ended question about what motivated them to take the course, whether they worked on the robotics activities outside of class, and did they think robotics activities are and will be useful?

In addition, participants were given an opinion questionnaire to fill out individually after the end of the workshops.

### 3.3. Implement of the Research

Following the proposed steps (Figure 1), the research was first carried out by making an initial diagnosis. Therefore, a literature revision was conducted on the interests of the female gender (see Table 1).

**Table 1.** Interests of the female gender.

| Source | Interests |
|---|---|
| Michael et al. [71] | The girls showed interest in the use of tangible interfaces. |
| Su et al. [72] | The women showed interest in artistic areas. |
| Yamtinah et al. [73] Makarova et al. [74] | Women showed interest in helping people. In STEM areas, women lost interest very quickly in the areas of math and physics. |
| | Women are more interested in areas of health. |
| Negrini et al. [75] | Robots tend to be of greater interest to boys, due to their greater interest in technical skills. |
| Shaqiri et al. [76] | Differences in visual perception, are usually more visual. |

According to the identified interests of women, it is proposed to design a set of workshops using Arduino hardware as a platform. Therefore, in the planning workshops, the contents are designed with the aim of combining theory and practice. The theory is developed for 30 min and the rest of the time is practice. Each workshop has a duration of 2 h. The set of workshops has the objective of introducing basic concepts of educational robotics, in the areas of electronics and programming.

Electronics is one of the basic areas for the development of robots. It is the main source of robots to perceive and react according to the environment. Therefore, it includes basic elements to perceive, send, and process signals from/to the different sensors and robotic actuators. The content defined for this discipline is described in Table 2.

**Table 2.** Thematic fields of the discipline "electronics" associated with learning level.

| | | |
|---|---|---|
| 1. Basic Components | 1.1. Definition of Microcontrollers (Arduino) | basic |
| | 1.2. What is a circuit? (Voltage and electricity) | basic |
| | 1.3. Leds | |
| | 1.4. Resistors | basic |
| | 1.5. Jumpler Cables | basic |
| | 1.6. How to make assemblies with a Protoboard? | basic |
| 2. Sensors (inputs) | 2.1. definition perception | basic |
| | 2.2. Light based sensors | basic |
| | 2.3. Temperature | basic |
| | 2.4. Humidity | basic |
| | 2.5. Pressure | |
| 3. Responses (outputs, Actuators, electronic components) | 3.1. Visual feedback | basic |
| | 3.2. Audio feedback | |

Computing is also one of the most important fields that form the basis of robotics. Therefore, it includes the process of designing computer programs. The contents defined for this discipline are in Table 3.

**Table 3.** Thematic fields of the discipline "programming" associated with learning level.

| | | |
|---|---|---|
| 5. Programming concepts | 5.1. What is programming? | basic |
| | 5.2. Basic Aspects (data types, variables and propositional logic) | basic |
| 6. Control statements | 6.1. Conditional sentences | basic |
| | 6.2. Repetitive sentences | basic |
| | 6.3. Variables | basic |
| | 6.4. Operators | basic |

Five workshops were proposed (See Table 4): (1) Creating interactive stories with Scratch; (2) medical science and electronics; (3) interactive toys; (4) music and electronics; (5) smart planter. For each workshop, a primer-type digital material was created, which provides information on concepts and can also be used by the teachers themselves to transmit their knowledge to the students. Equally, a low-cost kit was offered, to make it more accessible.

**Table 4.** Workshops associated with STEM 21st Century Skills.

| Workshop | Description | Skills | Tool(s) |
|---|---|---|---|
| Interactive stories | Introduction to programming concepts through storytelling. | Technology use, problem solving | Scratch |
| Medical sciences and electronics | Introduction to electronics concepts, such as: protoboard, Arduino UNO, Jumper cables, Leds diode and resistors. | Technology use, Problem solving, and creativity | TinkerCad Arduino |
| Interactive toy | Using basic electronics components to build a face that can simulate reactions or emotions. Visual, tactile, and auditory responses are worked on. | Technology use, Problem solving, and creativity | TinkerCad Arduino |
| Music and electronics | Introduction to music concepts and piezoelectric sensor. | Technology use, Problem solving, and creativity | TinkerCad Arduino |
| Smart planter | Using temperature and humidity sensors to build a low-cost smart plant monitor. | Technology use, Problem solving, and creativity | TinkerCad Arduino |

To carry out these five workshops, the kit included: Arduino UNO with a USB cable and a 400-point breadboard, (2) red, green, and blue LED diodes, (10) 220 Ohm resistors, (3) 10 Kohm resistors, (1) an LM35 temperature sensor, 1 DHT11 temperature, and humidity sensor module, a hygrometer sensor, a KOhm potentiometer, (2) LDR photoresistors, (5) 27 mm piezoelectric, a buzzer, and a male–female and male–male jumper cable pack. As such, the kit was priced at $30.

Before starting the workshops, instructional material was developed on how to enroll or register on virtual platforms such as Scratch [77] and Tinkercard, since there were schoolteachers who did not have much contact with technology. Moreover, as a way to approach the task, the Flipgrid platform enabled them to make a short video to introduce themselves and their interest in doing the workshops.

Below is a brief description of each workshop:

### 3.3.1. Workshop 1: Interactive Stories

The aim was to introduce programming concepts using the Scratch platform. Therefore, a digital material was designed (See Figure 2), creating a character called RoboTIC, in which a story could be designed, and the different programming blocks could be taught. It was also explained to them that the creation of stories helps to develop competencies related to creativity [78] and abstraction [79].

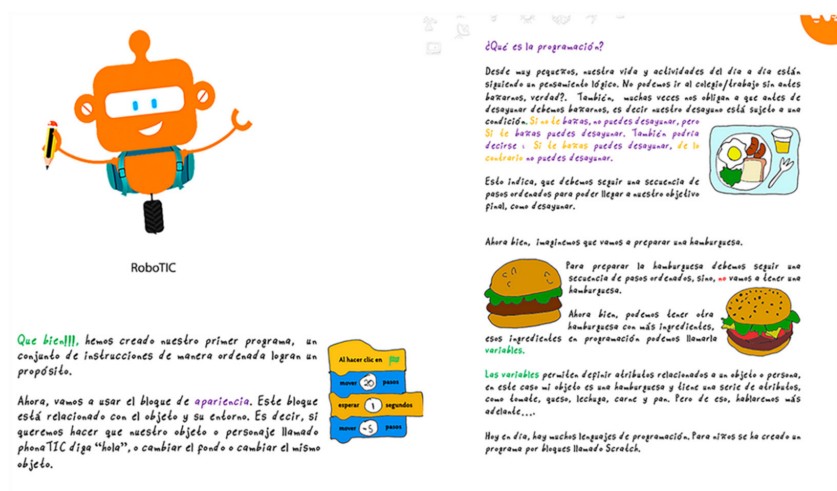

**Figure 2.** Design of Workshop 1 material (Spanish version).

### 3.3.2. Workshop 2: Medical Sciences and Electronics

In the second workshop, basic concepts of electronics were introduced. Digital material was designed to teach concepts related to electronics, such as: What is an electrical circuit? What is a diode? What is voltage? How is a breadboard used? What is a resistor? Why should the resistor be used together with the Led diode? Finally, a brief concept was introduced: What is a sensor?

They were also introduced to medical science concepts, for example simulating physiological responses (heart rate) or capturing physiological responses (skin temperature or conductivity).

Before carrying out a physical assembly, the simulation was carried out supported by the Tinkercard platform. Once the assembly and programming were working correctly in the simulation, the code was exported to Arduino. The physical assembly would then be carried out using the basic components: LED diode, resistor, and LM35 (See Figure 3).

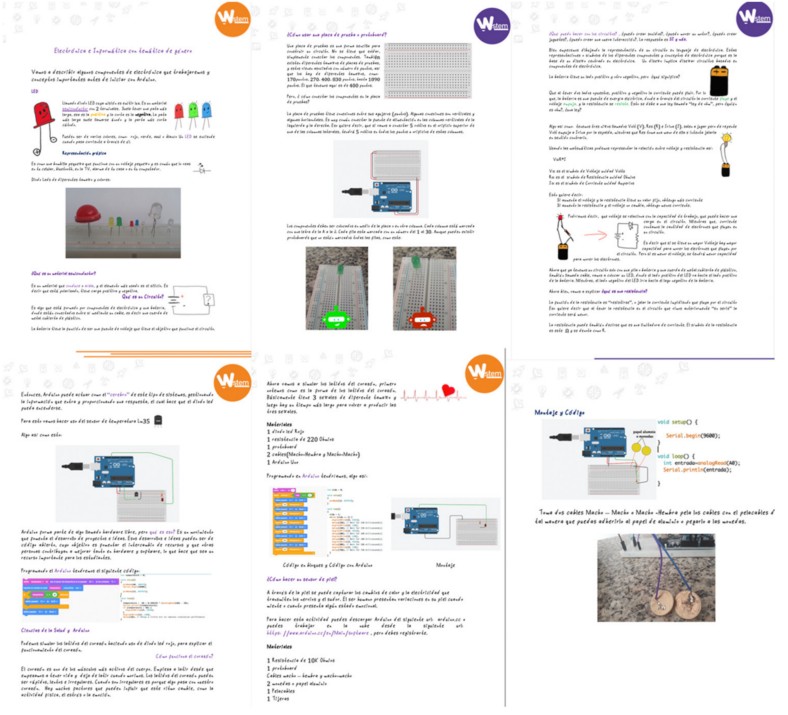

**Figure 3.** Design of Workshop 2 material (Spanish version).

From this workshop, explanatory videos were made and the whole process was carried out (Figure 4), which consisted of (1) assembly of the circuit using TinkerCard; (2) programming the circuit using Tinkercard; (3) simulation of operation; (4) exporting the code to Arduino; (5) physical assembly; (6) compiling and loading the program in Arduino; and (7) testing the operation in the physical assembly.

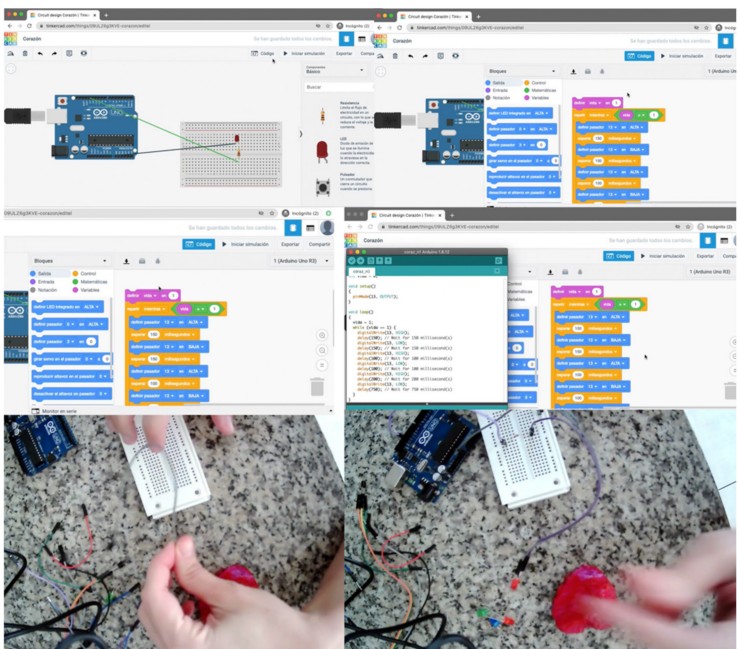

**Figure 4.** Explanatory video of the "heart rhythm simulation" activity.

### 3.3.3. Workshop 3: Interactive Toy

In the third workshop, they learn about other electronic components such as variable resistors, the potentiometer, and photo-resistance (See Figure 5). It was also referenced that one can have components that act as outputs or responses to an event, and there are other components that are called sensors that capture physical signals.

Therefore, in this workshop, it was decided to build an interactive face that can react to certain responses. For the construction of the interactive face, facial features and electronic components were selected to capture signals and react to those signals. Therefore, the characteristics that were used were the eyes (LED diodes), the nose (potentiometer), the mouth (buzzer), and the cheeks (Photo-resistances). The interaction was that every time his nose was pinched, he would react to it as if it were painful, so he would have a reaction of blinking his eyes and complaining through the noise that the buzzer produces. In turn, when the cheeks were touched, it would produce another more pleasant reaction using only the eyes but not the mouth.

We performed this workshop so that children, for example with special needs or younger children, could be taught various emotional responses. The activity was therefore called "expressing facial emotions". As a task, they were asked to design a face from recyclable cardboard and place the electronic components where they believed it was more convenient to express one or more emotions.

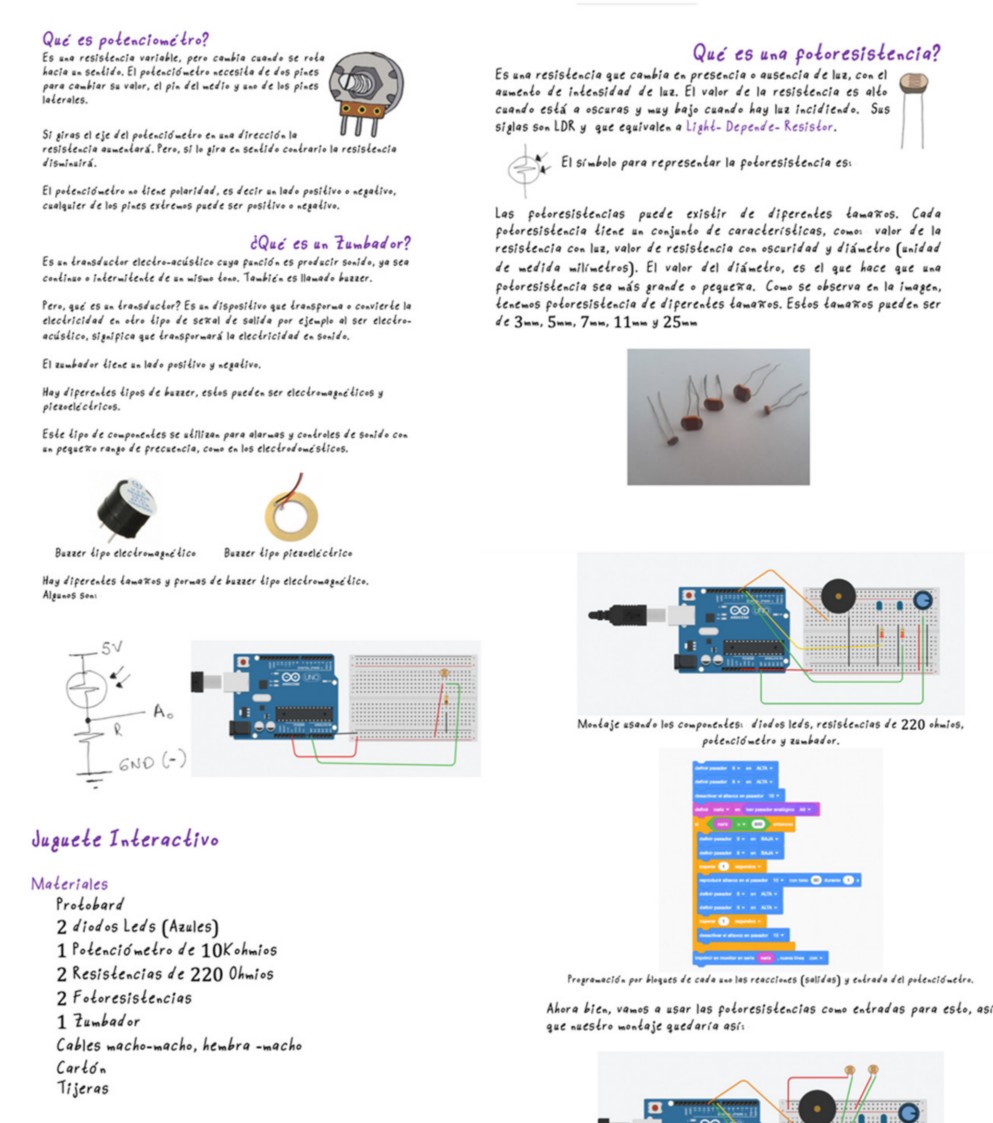

**Figure 5.** Design of Workshop 3 material (Spanish version).

### 3.3.4. Workshop 4: Music and Electronics

In the fourth workshop, they learned to use piezoelectric together with the buzzer. Therefore, it was decided to make a musical instrument with recyclable material, where each key is a piezoelectric and the sound is produced through the buzzer (see Figure 6). The idea of this workshop was to introduce them to art through music. Therefore, a brief introduction was given on how music can benefit children.

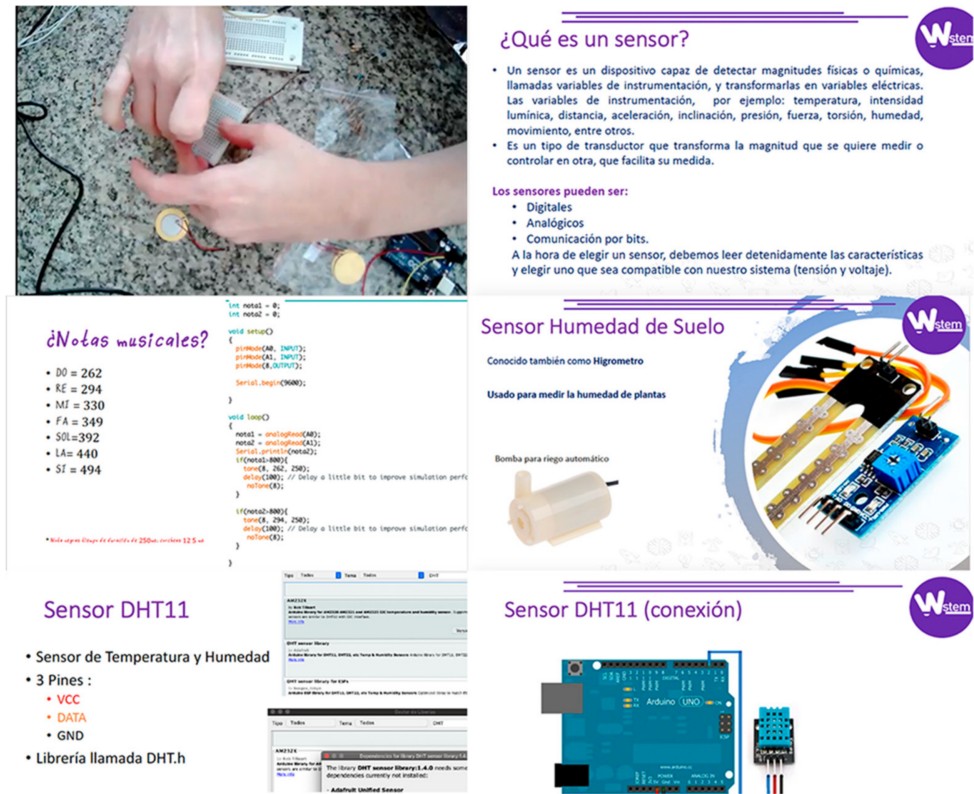

**Figure 6.** Design of Workshop 4 and 5 materials (Spanish version).

3.3.5. Workshop 5: Smart Planter

This delves deeper into the concepts of sensors (see Figure 6), and how you can build a planter that can emit alerts if a plant needs water, or the temperature is very high, and it needs water. Therefore, LEDs or buzzer diodes were used so that the system was able to react.

From this workshop, explanatory videos were made and the whole process was carried out (Figure 4), which consisted of: (1) Assembly of the circuit using Tinkercard; (2) programming the circuit using Tinkercard; (3) simulation of the operation; (4) exporting the code to Arduino; (5) physical assembly; (6) compiling and loading the program in Arduino; (7) testing the operation in the physical assembly.

The following 5E model was applied for each workshop:

*Engagement:* Student's prior knowledge about programming concepts, introduction to a microcontroller with Arduino UNO, and electronics components. The teacher starts the workshops with a question to solve a specific problem. The students produce ideas using brainstorming, and these ideas create a discussion environment in class.

*Exploration:* Each participant is provided with learning material. The teacher introduces these materials to the students and provides information about their use. The students create an algorithm according to the workshop. For example, in Workshop 1, students must create a story using the Scratch platform. The teacher guided the coding on the Scratch coding platform based on this algorithm.

From the second workshop, the teacher guided the coding on TinkerCad and then students made the connection using the breadboard and uploaded the development code to the Arduino card.

*Explanation:* The teacher creates a classroom discussion environment by asking students about programming and electronics according to the context of the workshop.

*Elaboration:* Students must carry out the assembly using the Arduino Uno by agreeing on what they have worked on with the TinkerCad platform and the solution to the problem.

*Evaluation:* The students must carry out the assembly according to the workshop to be solved. Each task consisted of uploading a very short 1 to 2 min video where they shared how it works through Arduino and electronic components (See Figure 7). For example, in Workshop 1, participants were asked to "create your own story" where they were given graphic material so that they could create their interactive story using the Scratch platform.

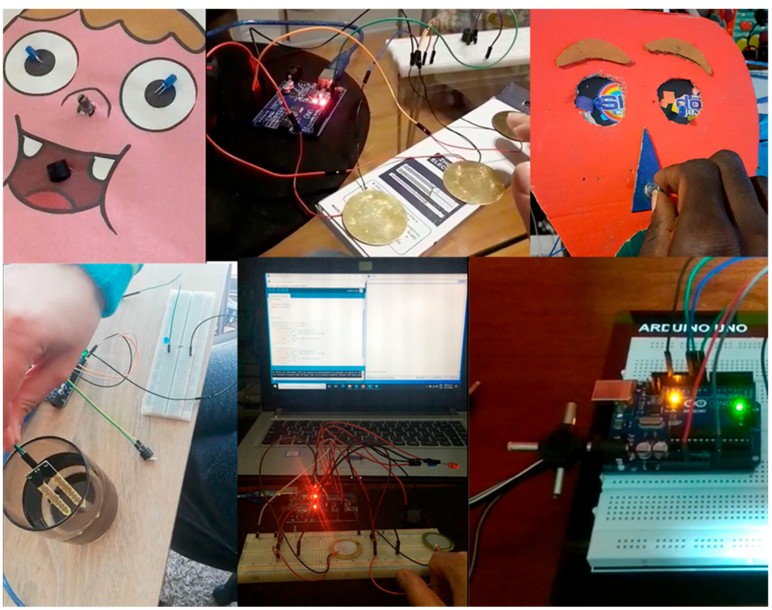

**Figure 7.** Some activities handed in by the participants.

## 4. Data Analysis

In the research, quantitative data obtained were demographic data and robotic attitude. The first workshop was held synchronously with 122 participants, and the following 4 workshops had an average of 57 attendees. Attendance was distributed as follows: 50 people attended all 5 sessions, 17 attended 4 of them, 15 attended 3, 12 attended 2, and 200 attended 1 or less. Each week they were assigned a task related to the workshop. A synchronous space was also created to attend to doubts or technical problems.

According to the demographic information of the participants, 52.5% were women and 47.5% were men. Furthermore, 55.2% were schoolteachers, 23.7% were students, 8.7% were higher education teachers, 9% were professionals, and 3.3% were other. Some answers related to motivation and the robotics activities were: "interest in learning about robotics", "teach electronics better to my students", "learn to program with Arduino Uno", "learn more robotics for my classes", "learn more about the scope of the Arduino", "I am a physics teacher and I would love to use Arduino in my classes", "learn robotics to incorporate it into teaching-learning", "learn about new technologies", "develop computational thinking", "learn to program sensors", "improve my robotics skills for my students", and "learn to practice it with preschool children". Therefore, in each of the workshops, there were teachers who knew how to use the technology, as well as others who did not.

Attitudes towards technology were evaluated through aspects such as interest (12 items) and curiosity (8 items) (see Table 5). Moreover, the word "robots" was changed to "electronics and programming". The items were evaluated by a Likert-like scale such as "NO!", "no", "neither yes or no", "yes" and "YES!" [80], which was scored with 1 to 5 scoring where 1 was "NO!" and 5 was "YES!". Items in the interest aspect included computers are interesting to me; I use the internet to find information about computers; I try to do activities related to computers; I like to explore computers; I feel good when I learn about computers, and I have a good feeling about computers. Meanwhile, items evaluated by the curiosity aspect where I am interested in discovering things about computers; I get excited about discussing computers; It is cool to learn new things about robots; I enjoy

exploring new ideas about computers; I am often trying to find out more about computers. Participants responded 80% "YES!" and 20% "yes" for both interest and curiosity aspects. Some of the teachers had knowledge of educational robotics, and they had already interacted with Arduino and Scratch, while others did not know either. Moreover, most of them were technology teachers.

**Table 5.** Attitudes towards technology. Taken from [70].

| Sub-Scale | Item |
|---|---|
| Interest | 1. I would like to learn more about robots. |
| | 2. Computers are interesting to me. |
| | 3. Topics like robots just don't grab my interest. |
| | 4. Robots are interesting to me. |
| | 5. I use the Internet to find information about computers. |
| | 6. I like to watch TV shows and/or read about robots. |
| | 7. I try to do activities related to computers. |
| | 8. I like to explore computers. |
| | 9. I like to do robotics activities. |
| | 10. I feel good when I learn about computers. |
| | 11. Robots are boring to me. |
| | 12. I have a good feeling about computers. |
| Curiosity | 1. I am curious about robots. |
| | 2. I am interested in discovering things about computers. |
| | 3. I get excited about discussing computers. |
| | 4. It is cool to learn new things about robots. |
| | 5. I enjoy exploring new ideas about computers. |
| | 6. I look for as much information as I can about robots. |
| | 7. Everywhere I go, I am out looking for new things about robots. |
| | 8. I am often trying to find out more about computers. |

At the end of the last workshop, a survey was conducted to receive feedback on the experience of the course. Eighteen participants responded. Some of the questions that were asked were: What was your previous experience in programming and electronics? How do you rate your learning experience of the course? Do you think the proposed activities are attractive for girls and young women? Which workshop was the one that you liked the most? What was the workshop that you liked the least? How do you rate your level of commitment in the course?

Some of the responses obtained were as follows. The workshop that they liked the most was workshop 4 (33.3%). The one they liked the least was their first introductory workshop on programming with Scratch (38.9%). In turn, they would have liked to go deeper into workshop 4. Of the participants, 27.8% had no experience with programming or electronics. Moreover, 55.6% considered that the proposed activities were attractive to girls and young women.

Some of the observations that were obtained were: "the course should have been longer and deepened more in concepts such as scratch", "I thought it was excellent", "I would like the next intermediate level version of the course", "I liked it a lot, I had Zero approach to Arduino. Sometimes I was behind because I had no knowledge and needed support", "the course seemed very didactic to me, an important potential. I would have liked more time for practical activities", and "I really liked it because it can be applied in all professions".

When the participants had interacted with the physical breadboard, we applied a short survey to determine their experience with the use of the breadboard and Arduino. Of the 52 participants who attended remote classes, 2% found the experience "very bad", 4% "bad", 25% "normal", 40% "good", and 29% "very good". Another question asked what they had had the most difficulty with when interacting with Arduino. In total, 23% answered they had understood the concepts, 25% did not have the materials, 8% did not work with Arduino, and 62% did not have much time to do the assigned tasks.

We obtained other open answers to how they would apply the knowledge: "develop play and experimentation activities with high school students", "apply it to students of 1 and 2 means to use different sensors", "the medical science one to explain the behavior of the human body, interactive toy as a toy robot, intelligent planter as a greenhouse, that is, they can be applied at all educational levels"," I am a teacher of preschool education in the classroom. I would apply Scratch as a work tool".

## 5. Discussion and Conclusions

Learning Arduino involves learning many technologies that allow for developing different projects for different needs. Therefore, the set of workshops introduced hardware elements commonly used in robotics fields, such as actuators, sensors, control boards, and outputs (lights, sound). This type of project allows participants to produce ideas by integrating electronics and programming. These types of ideas can be related to the environment, traffic, energy, recycling, health, and safety; and the ideas produced can be different and design unusual applications.

Therefore, Arduino is an alternative to introduce programming and electronics concepts. It is easy to integrate hardware and mechanical components for application development. However, for younger ages, it requires more attention from the instructor to the students. In addition, it was observed that participants learn to strive to produce ideas to find solutions to the problems they observe by making use of robotics mechanisms. Therefore, it can help improve students' creativity by enabling them to think differently and critically.

The first workshop, "creating storytelling" with Scratch, enables the programmer to become creative. Create a story includes more than one character and scene, which allows one to program dialogs and actions. For example, the user may include audio sounds or time sequences or may construct a message to be passed among different sprites. Digital storytelling is a process of designing and programming digital stories, wherein students can develop computational thinking skills and other skills such as digital literacy and problem-solving skills [81]. There are key elements of the digital story such as the setting of the story, characters, scenes, sequence of events, and narrative [82]. Therefore, making interactive stories with Scratch can help develop a set of skills not only focused on programming.

The application of the methodological approach allowed the design of workshops focused on the needs of a specific user and context. Therefore, the proposal is a process that includes methodological guidelines to apply a robotics curriculum for the implementation of ER through the development of projects with a gender approach.

Zint [83] mentioned that attitudes are learnable and teachable. Therefore, interest and curiosity lead to the development of positive emotions. Therefore, the present research showed greater participation of the female gender and had positive acceptance and impact. The interest of the participants in learning and applying this with their students was thus reflected. Teachers already know the importance of the use of Arduino coding applications integrated into the 5E learning in STEM teaching in improving students' attitudes toward electronic and programming in an interdisciplinary way and from an early age in students. In the workshops, students used engineering concepts to assemble electronic devices and arrive at the solution to the problem. In this way, students are exposed to situations that allow them to generate new ideas and create new algorithms.

However, even schools do not have clear educational policies towards the use of educational robotics to include the topic within the academic curriculum.

## 6. Limitations

This study has several limitations. The limitations of the present study were (1) there was no control group, so the proposed phases could be validated; (2) there was a wide variety of teachers belonging to private and public schools, so the profile of the participants was not uniform; (3) not all participants performed the assembly with the use of the kit. In addition, it was a zoom course, which did not allow us to see the progress in a certain way.

Lastly, (4) only the Arduino microcontroller card and its basic components are used as it is easy to use and cheap.

**Funding:** This research received no external funding.

**Conflicts of Interest:** The authors declare no conflict of interest.

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
