# Peer review of "A Methodological Approach to the Teaching STEM Skills in Latin America through Educational Robotics for School Teachers"

_electronics, doi:10.3390/electronics11030395_

Round 1
Reviewer 1 Report
The aim of the paper was to present some results regarding an Erasmus+ project focused on the development and training of an "educational robotics course with a gender perspective",
The target group was "290 people from different countries registered: Chile (124), Colombia (98), Ecuador (11), Mexico (26), Costa Rica (3), Peru (23) and Europe (5), where 47.5% of the registered 268 correspond to men and 52.5% to women."
Relevance: The paper can be (more or less) relevant for people interested in a specific case study (i.e. Teaching STEM skills in Latin America). The results are statistical rather than scientific...
Significance: The paper is “fairly significant” due to limitation of used concepts, i.e. only some statistical results based on questionnaire.
Quality: The manuscript is technically correct. The technical matters are clearly explained and no contradictions were found.
On the other hand, the theoretical aspects are (generally speaking) truisms and the used techniques are notorious.
Clarity: The entire manuscript is well written but, in my opinion, the paper (with the premises / conclusions) the paper is a dissemination action rather than a scientific analysis
Author Response
The manuscript was updated.

Reviewer 2 Report
My comments:
- The topic of this paper is interesting and it will make contributions in related research field.
- The “2.3. Methodologies”, “3. Course Design”, and “4. Discussion” are too brief, they should be reinforced to describe in more detail.
- The section “Conclusions and Future Work” must be reinforced more. For example, the contributions to academic research as well as theoretical implications, research limitations, and suggestions for further research.
Author Response
The manuscript was updated.

Reviewer 3 Report
Dear Authors,
Pls help to revise as below suggestions:
Abstract
The background, aims, objectives, the scope of works of the study is not described clearly.
How does the participation of the female gender have a direct impact on their students? In what ways?
Evidence is required to support the literature.
The variables are not clearly presented and identified in the abstract.
The research method is missing.
In general, the abstract is confusing and presented illogically. The author is required to rewrite the abstract.
What is the significance and contribution of this study?
The current format is like an advertisement or promotion of the introductory course more than an abstract of a research paper.
Introduction
The introduction is generally acceptable with the background and investigation of the whole study.
- Line 39: “…United States and South Korea, girl’s education is considered an essential element for economic development” Any other countries from Europe and Asia for example?
- Line 66:” … Educational Robotics is a way to increase student interest in science and engineering, especially women” Not a clear definition of Educational Robotics. Authors are advised to clearly identify and define these robotics.
- Line 73:” …Authors identified that one reason female students may be less confident in their technical and building skills is that females’ students may simply have less experience with building,…” Who are the authors? It is necessary to clearly explain.
- Line 96:” Most papers examined make use of robotics kits, where students must use a programming language to code the robots…” The authors are advised to clearly identify most papers.
- Line 99:” Some questions in this study are: What aspects should be considered to motivate women to choose studies related to engineering, especially computer sciences and electronics? How could a gender-sensitive introductory course on educational robotics…? Are they research hypotheses? Aim? Objectives?
Literature Review
- Background: Do the authors consider briefly introducing the history/ development/ background of Educational robotics (ER)?
- Line 140: “Today, ER is being implemented in schools as an alternative to empower students…” Any literature to support for reference?
- Line 151: “this study they carried out three activities. The first activity was to code the robots.” Any other words to replace the word “activities”? This paper should be a research paper and not activities promotion advertising.
- Line 156-157: 2.2. Gender in Educational Robotics “Children tend to be more motivated by the use of technology. However, Sapounidis et al. [33] found that girls..” How about any comments on the boys for comparison?
- This is very important to find out why the girls do not have a strong interest in the topic of science/ mathematics such as Educational robotics (ER) from the literature. Is the poor performance such as poor academic results of the girls on the subjects make them no interest? Does past academic performance/experience constitute a successful element for the girls to have any interest in those subjects?
Methodology
- Line168: 2.3. A new section called “Research Methodologies” is required.
- Line 237: 3. Course Design. A new section called “Methodologies” is required. Pls. help to clarify.
- From Line 169 – 229, this is a bit like a literature review of the methods of application and operation of Educational robotics (ER) by other schools or researchers more than an independent section of the Research Methodologies of this paper. Pls, consider rewriting the Research Methodologies.
- Line 256: To promote the course, advertising banners were designed (see Figure 1) where it was promoted through social networks such as Facebook, Instagram and sending emails schools, mainly in Chile.” It is necessary to indicate the period.
- Line 270 - 284: 3.1. Participants. Pls consider that the data should be in the session of data
- Line 330 - 337: 3.2. Workshops. Pls consider that the data should be in the session of data
- Line 381 - 405: 3.2. Workshops. Pls, consider that the data should be in the session of data analysis.
- The current format of research method and data analysis is rather confusing.
Results and Discussion
- It is necessary to add a new section of data analysis for the research paper.
- Line 411: “The course was widely accepted by school teachers, especially female teachers. Of the…” This seems like a course promotion more than a research ER project. This is the main problem of the current format of this “Research paper”.
- It is advised that the authors have to consider discussing how to improve the feedback and awareness of the ER from the point of view of female respondents from the study.
- Line 416 - 421: This is just a literature review again, but not the own discussion materials suggested by the authors. Pls, consider rewriting this part of the discussion.
- Line 422 - 429: The focus of this paragraph is vague. It is suggested that the authors should have a clear theme in each paragraph.
- Line 430 - 445: Again, the paragraph is like the literature review. The authors are advised to review and rewrite the discussion.
- In the discussion, the authors are advised to consider rewriting and put more own analysis and suggestions after conducting the on-site experiments from the participants.
Conclusions
- No clear recommendation
- No clear limitation
References
Sufficient number of literature review
Author Response
The manuscript was updated.

Reviewer 4 Report
It is an interesting topic with potential contributions to exploring the use of educational robotics in preparing pre-service teachers. However, the authors are over ambitious to integrate many concepts in this study. The study has some major flaws. The research questions are vague, which need to be reframed. The first half of the literature is with many assumptions, not based on existing literature or research data. This paper also exists major misalignment issue among research questions, study design and results.
Author Response
The manuscript was updated.

Round 2
Reviewer 1 Report
After revision, my opinion is that the manuscript can be published in its current form
Reviewer 2 Report
This paper is qualified to be published.